# Generative Feature Training of Thin 2-Layer Networks

**Johannes Hertrich**                                     *johannes.hertrich@dauphine.psl.eu*
*Université Paris Dauphine - PSL*

**Sebastian Neumayer**                                  *sebastian.neumayer@math.tu-chemnitz.de*
*Technische Universität Chemnitz*

**Reviewed on OpenReview:** *https://openreview.net/forum?id=6oXNpKuBDK*

## Abstract

We consider the approximation of functions by 2-layer neural networks with only a few hidden weights based on the squared loss and small datasets. Due to the highly non-convex energy landscape, gradient-based training often suffers from local minima. As a remedy, we initialize the hidden weights with samples from a learned proposal distribution, which we parameterize as a deep generative model. To train this model, we exploit the fact that with fixed hidden weights, the optimal output weights solve a linear equation. After learning the generative model, we refine a set of sampled weights with a gradient-based feature refinement in the latent space. Here, we also include a regularization scheme to counteract potential noise. Finally, we demonstrate the effectiveness of our approach by numerical examples.

## 1 Introduction

We investigate the approximation of real-valued functions $f \colon [0,1]^d \to \mathbb{R}$ based on samples $(x_k, y_k)_{k=1}^M$, where $x_k \in [0,1]^d$ are independently drawn from some distribution $\nu_{\text{data}}$ and $y_k \approx f(x_k)$ are possibly noisy observations of $f(x_k)$. To achieve this, we study parametric architectures $f_{w,b} \colon [0,1]^d \to \mathbb{R}$ of the form

$$f_{w,b}(x) = \mathfrak{Re}\left( \sum_{l=1}^{N} b_l \Phi(\langle w_l, x \rangle) \right), \tag{1}$$

where $\mathfrak{Re}$ denotes the real part, $\Phi \colon \mathbb{R} \to \mathbb{C}$ is a nonlinear function, and $w_1, ..., w_N \in \mathbb{R}^d$ are the features with corresponding weights $b_1, ..., b_N \in \mathbb{C}$. If the function $\Phi$ is real-valued, the model (1) simplifies to a standard 2-layer neural network architecture without $\mathfrak{Re}$ and with $b_1, ..., b_N \in \mathbb{R}$. The more general model (1) also covers other frameworks such as random Fourier features (Rahimi & Recht, 2007). Since the Pareto principle suggests that most real-world systems are driven by a few low-complexity interactions, we are interested in representations (1) with only a few features $w_l$. Such an explicit restriction of $N$ also mitigates overfitting, as seen in sparse neural networks, compressed sensing and feature selection.

For fixed $\Phi$ and $N$, we aim to find $(w, b) \in \mathbb{R}^{d,N} \times \mathbb{C}^N$ such that the $f_{w,b}$ from (1) approximates $f$ well. From a theoretical perspective, we can obtain such $(\hat{w}, \hat{b})$ by minimizing the mean squared error (MSE), namely

$$(\hat{w}, \hat{b}) \in \operatorname*{arg\,min}_{w,b} \| f - f_{w,b} \|_{L^2(\nu_{\text{data}})}^2. \tag{2}$$

In practice, we do not have access to $\nu_{\text{data}}$ and $f$, but only to data points $(x_k, y_k)_{k=1}^M$, where $x_k$ are iid samples from $\nu_{\text{data}}$ and $y_k$ are noisy versions of $f(x_k)$. Hence, we replace (2) by the empirical risk minimization

$$(\hat{w}, \hat{b}) \in \operatorname*{arg\,min}_{w,b} \sum_{k=1}^{M} |y_k - f_{w,b}(x_k)|^2. \tag{3}$$

However, if $M$ is small, minimizing (3) can lead to overfitting towards the training samples $(x_k, y_k)_{k=1}^M$ and poor generalization. To address this issue, we investigate the following principles.

- We use $f_{w,b}$ of the form (1) with small $N$. This amounts to the implicit assumption that $f$ can be *sparsely* represented using (1). Unfortunately, under-parameterized networks ($N \ll M$) are difficult to train with conventional gradient-based algorithms (Boob et al., 2022; Holzmüller & Steinwart, 2022), see also Table 1. Hence, we require an alternative training strategy.

- Often, we have prior information about the regularity of $f$, i.e., that $f$ is in some Banach space $\mathcal{B}$ with a norm of the form

$$\|f\|_{\mathcal{B}}^p = \int_{[0,1]^d} \|Df(x)\|_q^p \mathrm{d}x, \tag{4}$$

  where $D$ is some differential operator and $p, q \geq 1$. A common example within this framework is the space of bounded variation (Ambrosio et al., 2000), which informally corresponds to the choice $D = \nabla$, $q = 2$ and $p = 1$. In practice, the integral in (4) is often approximated using Monte Carlo methods with uniformly distributed samples $(\tilde{x}_m)_{m=1}^S \subset [0,1]^d$. If we use (4) as regularizer for $f_{w,b}$, the generalization error can be analyzed in Barron spaces (Li et al., 2022).

**Contribution**   We propose a generative modeling approach to solve (3). To this end, we first observe that the minimization with respect to $b$ is a linear least squares problem. Hence, we can analytically express the optimal $\hat{b}$ in terms of $w$, which leads to a reduced problem. Using the implicit function theorem, we compute $\nabla_w \hat{b}(w)$ and hence the gradient of the reduced objective. To facilitate its optimization, we replace the deterministic features $w$ with stochastic ones, and optimize over their underlying distribution $p_w$ instead. We parameterize this distribution as $p_w = G_{\theta\#}\mathcal{N}(0, I_d)$ with a deep network $G_\theta \colon \mathbb{R}^d \to \mathbb{R}^d$. Hence, we coin our approach as *generative feature training*. Further, we propose to add a Monte Carlo approximation of the norm (4) to the reduced objective. With this regularization, we aim to prevent overfitting.

## 2   Related Work

**Random Features**   Random feature models (RFM) first appeared in the context of kernel approximation (Rahimi & Recht, 2007; Liu et al., 2021), which enables the fast computation of large kernel sums with certain error bounds, see also Rahimi & Recht (2008); Cortes et al. (2010); Rudi & Rosasco (2017). A similar strategy is pursued by Huang et al. (2006) under the name extreme learning machines. Sparse RFMs (Yen et al., 2014) of the form (1) with only a few active features can be computed based on $\ell_1$ basis pursuit (Hashemi et al., 2023). Since this often leads to suboptimal approximation accuracy, later works by Xie et al. (2022); Saha et al. (2023); Bai et al. (2024) instead proposed to apply pruning or hard-thresholding algorithms to reduce the size of $w$. Commonly, the features $w$ are sampled from Gaussian mixtures with diagonal covariances. Unlike our approach, all these methods begin with a large feature set that has to contain sufficiently many relevant ones. Towards this strong implicit assumption, Potts & Schmischke (2021); Potts & Weidensager (2025) propose to identify the relevant subspaces for the feature proposal based on the ANOVA decomposition. Unfortunately, this only works if the features $w$ itself are sparse (few non-zero entries), see Figure 1. Sparse features also enable the fast evaluation of the $f_{w,b}$ from (1) via the non-equispaced fast Fourier transform (Dutt & Rokhlin, 1993; Potts et al., 2001). For kernel approximations, this can be also achieved with slicing methods (Hertrich, 2024; Hertrich et al., 2025), which are again closely related to RFMs (Rux et al., 2025).

**Adaptive Features**   Besides our work, there are several attempts to design data-adapted proposal distributions $p_w$ for random features (Li et al., 2019c; Dunbar et al., 2025). Recently, Bolager et al. (2023) proposed to sample the features $w$ in regions where it matters, i.e., based on the available gradient information. While this allows some adaption, the $w$ still remain fixed after sampling them (a so-called greedy approach). Towards fully adaptive (Fourier) features $w$, Li et al. (2019b) propose to alternately solve for the optimal $b$, and to then perform a gradient update for the $w$. Kammonen et al. (2020) propose to instead update the $w$ based on a Markov Chain Monte Carlo method. Unlike our approach, both methods do not incorporate the gradient information of $b$ into the update process of $p_w$. It is well known that the surrogate alternating updates may perform poorly in certain cases. Note that learnable features have been also used in the context of positional encoding (Li et al., 2021) and implicit kernel learning (Li et al., 2019a).

**2-Layer ReLU Networks**   We can interpret 2-layer neural networks as adaptive kernel methods (E et al., 2019). Moreover, they have essentially the same generalization error as the RFM. Several works investigate the learning of the architecture (1) with $\Phi = \mathrm{ReLU}$ based on a (modified) version of the empirical risk minimization (3). Based on convex duality, Pilanci & Ergen (2020) derive a semi-definite program to find a global minimizer of (3). A huge drawback is that this method scales exponentially in the dimension $d$. Later, several accelerations based on convex optimization algorithms have been proposed (Mishkin et al., 2022; Bai et al., 2023). Following a different approach, Barbu (2023) proposed to use an alternating minimization over the parameters $w$ and $b$ that keeps the activation pattern fixed throughout the training. While this has an improved complexity of $\mathcal{O}(d^3)$ in $d$, the approach is still restricted to ReLU-like functions $\Phi$. Moreover, gradient-based optimization of the parameters for a generative (proposal) network such as ours is empirically known to scale very well with $d$. A discussion of the rich literature on global minimization guarantees in the over-parameterized regime ($N \gg M$) is not within the scope of a sparse architecture (1).

**Bayesian Networks**   Another approach that samples neural network weights is Bayesian neural networks (BNNs) (Neal, 2012; Jospin et al., 2022). This allows to capture the uncertainty on the weights in over-parameterized architectures. A fundamental difference to our approach and RFMs is that we sample the features $(w_l)_{l=1}^N$ independently from the same distribution, while BNNs usually learn a separate one for each $w_l$. Further, BNNs are usually trained by minimizing an evidence lower bound instead of (8), see for example (Graves, 2011; Blundell et al., 2015), which is required to prevent collapsing distributions.

## 3   Generative Feature Learning

Given data points $(x_k, y_k)_{k=1}^M$ with $y_k \approx f(x_k)$ for some underlying $f : [0,1]^d \to \mathbb{R}$, we aim to find the optimal features $w = (w_l)_{l=1}^N \subset \mathbb{R}^d$ and weights $b \in \mathbb{C}^N$ such that $f_{w,b} \approx f$, where $f_{w,b}$ is defined in (1). Before we give our approach, we discuss two important instances of the nonlinearity $\Phi : \mathbb{R} \to \mathbb{C}$ from the literature.

- **Fourier Features**: The choice $\Phi(x) = \mathrm{e}^{2\pi\mathrm{i}x}$ is reasonable if the ground-truth function $f$ can be represented by few Fourier features, e.g., if it is smooth. As discussed in Section 2, the deployed features $w$ are commonly selected by randomized pruning algorithms.

- **2-Layer Neural Network**: For $\Phi : \mathbb{R} \to \mathbb{R}$, we can restrict ourselves to $b \in \mathbb{R}^N$. Common examples are the ReLU $\Phi(x) = \max(x, 0)$ and the sigmoid $\Phi(x) = \frac{\mathrm{e}^x}{1+\mathrm{e}^x}$. Then, $f_{w,b}$ corresponds to a 2-layer neural network (i.e., with one hidden layer). Using the so-called bias trick, we can include a bias into (1). That is, we use padded data-points $(x_k, 1) \in \mathbb{R}^{d+1}$ such that the last entry of the feature vectors $w_l \in \mathbb{R}^{d+1}$ can act as bias. Similarly, an output bias can be included.

In the following, we outline our procedure for optimizing the parameters $w$ and $b$ for a general $f_{w,b}$ of the form (1). First, we derive an analytic formula for the optimal weights $b$ in the empirical risk minimization (3) with fixed features $w$. Then, in the spirit of random Fourier features, we propose to sample the $w$ from a proposal distribution $p_w$, which we learn based on the generative modeling ansatz $p_w = G_{\theta \#} \mathcal{N}(0, I_d)$. As last step, we fine-tune the sampled features $w = G_\theta(z)$ by updating a set of sampled latent features $z$ with the Adam optimizer. In order to be able to deal with noisy function values $y_k \approx f(x_k)$, we can regularize the approximation $f_{w,b}$ during training. Our complete approach is summarized in Algorithm 1.

### 3.1   Computing the Optimal Weights

For fixed $w = (w_l)_{l=1}^N$, any optimal weights $b(w) \in \mathbb{C}^N$ for (3) solve the linear system

$$A_w^{\mathrm{T}} A_w b(w) = A_w^{\mathrm{T}} y, \tag{5}$$

where $y = (y_k)_{k=1}^M$ and $A_w = (\Phi(\langle x_k, w_l \rangle))_{k,l=1}^{N,M}$. In order to stabilize the numerical solution of (5), we deploy Tikhonov regularization with small regularization strength $\varepsilon > 0$. This resolves the potential rank deficiency of $A_w^{\mathrm{T}} A_w$ and we compute $b(w)$ as the unique solution of

$$(A_w^{\mathrm{T}} A_w + \varepsilon I) b(w) = A_w^{\mathrm{T}} y. \tag{6}$$

---

**Algorithm 1** GFT and GFT-r training procedures.

---

1: **Given:** data $(x_k, y_k)_{k=1}^M$, architecture $f_{w,b}$ as in (1), generator $G_\theta$, latent distribution $\eta$
2: **while** training $G_\theta$ **do**
3:     sample $N$ latent $z_l \sim \eta$ and set $w = G_\theta(z)$
4:     compute optimal $b(w)$ and $\nabla_w b(w)$ based on (6)
5:     compute $\nabla_\theta \mathcal{L}(\theta)$ or $\nabla_\theta \mathcal{L}_{\text{reg}}(\theta)$ with automatic differentiation
6:     perform Adam update for $\theta$
7: **if** GFT-r **then**
8:     **while** refining $w$ **do**
9:         set $w = G_\theta(z)$
10:         compute optimal $b(w)$ and $\nabla_w b(w)$ based on (6)
11:         compute $\nabla_z F(z)$ or $\nabla_z F_{\text{reg}}(z)$ with automatic differentiation
12:         perform Adam update for $z$
13: **Output:** features $w$ and optimal weights $b(w)$

---

For $\epsilon \to 0$, the solution of (6) converges to the minimal norm solution of (5). A key aspect of our approach is that we can compute $\nabla_w b(w)$ using the implicit function theorem. This requires solving a linear equation of the form (6) with a different right hand side. For small $N$, the most efficient approach for solving (6) is to use a LU decomposition, and to reuse the decomposition for the backward pass. This procedure is implemented in many automatic differentiation packages such as PyTorch, and no additional coding is required.

By inserting the solution $b(w)$ of (6) into the empirical loss (3), we obtain the reduced loss

$$L(w) = \sum_{k=1}^M |f(x_k) - f_{w,b(w)}(x_k)|^2. \tag{7}$$

Naively, we can try to minimize (7) directly via a gradient-based method (such as Adam with its default parameters) starting at some random initialization $w^0 = (w_l^0)_{l=1}^N \subset \mathbb{R}^d$. We refer to this as feature optimization (F-Opt). However, $L(w)$ is non-convex, and our comparisons in Section 4 reveal that feature optimization frequently gets stuck in local minima. Consequently, a good initialization $w^0$ is crucial if we want to minimize (7) with a gradient-based method. In the spirit of random Fourier features, we propose to initialize the $w$ as independent identically distributed (iid) samples from a proposal distribution $p_w$. To the best of our knowledge, current random Fourier feature methods all rely on a handcrafted $p_w$.

### 3.2 Learning the Proposal Distribution

Since the optimal $p_w$ is in general not expressible without knowledge of $f$, we aim to learn it from the available data $(x_k, y_k)_{k=1}^M$ based on a generative model. That is, we take a simple latent distribution $\eta$ (such as the normal distribution $\mathcal{N}(0, I_d)$) and make the parametric ansatz $p_w = G_{\theta\#}\eta$. Here, $G_\theta \colon \mathbb{R}^d \to \mathbb{R}^d$ is a fully connected neural network with parameters $\theta$ and $\#$ denotes the push-forward of $\eta$ under $G_\theta$. To optimize the parameters $\theta$ of the distribution $p_w = G_{\theta\#}\eta$, we minimize the expectation of the reduced loss (7) with iid features sampled from $G_{\theta\#}\eta$, namely the loss

$$\mathcal{L}(\theta) = \mathbb{E}_{w \sim (G_{\theta\#}\eta)^{\otimes N}}[L(w)] = \mathbb{E}_{z \sim \eta^{\otimes N}}[L(G_\theta(z))] = \mathbb{E}_{z \sim \eta^{\otimes N}}\left[\sum_{k=1}^M |f(x_k) - f_{G_\theta(z), b(G_\theta(z))}(x_k)|^2\right], \tag{8}$$

where the notation $\mu^{\otimes N}$ denotes $N$-times the product measure of $\mu$. We minimize the loss (8) by a stochastic gradient-based algorithm. That is, in each step, we sample one realization $z \sim \eta^{\otimes N}$ of the latent features to get an estimate for the expectation in (8). Then, we compute the gradient of the integrand with respect to $\theta$ for this specific $z$, and update $\theta$ with our chosen optimizer. In the following, we provide some intuition why this outperforms standard training approaches. In the early training phase, most of the sampled features $w$ do not fit to the data. Hence, they suffer from vanishing gradients and are updated only slowly. On the other hand, since the stochastic generator $G_{\theta\#}\eta$ leads to an evaluation of the objective $L(w)$ at many different

locations, we quickly gather gradient information for a large variety of features. In particular, always taking fresh samples from the iteratively updated proposal distribution $p_w$ helps to get rid of useless features.

### 3.3 Feature Refinement: Adam in the Latent Space

Once the feature distribution $p_w = G_{\theta\#}\eta$ is learned, we sample a collection $z^0 = (z_l^0)_{l=1}^N$ of iid latent features $z_l^0 \sim \eta$. By design, the associated features $w^0 = G_\theta(z^0)$ (with $G_\theta$ being applied elementwise to $z_1^0, ..., z_N^0$) serve as an estimate for a minimizer of (7). Since these $w^0$ are only an estimate, we refine them similarly as described for the plain feature optimization approach from Section 3.1. More precisely, starting in $z^0$ instead of a random initialization, we minimize the function

$$F(z) = L(G_\theta(z)) = \sum_{k=1}^M |f(x_k) - f_{G_\theta(z),b(G_\theta(z))}(x_k)|^2, \tag{9}$$

where $L$ is the loss function from (7). By noting that $\nabla F(z) = \nabla G_\theta(z)^{\mathrm{T}} \nabla L(G_\theta(z))$, this corresponds to initializing the Adam optimizer for the function $L(w)$ with $w^0 = G_\theta(z^0)$, and to additionally precondition it by the Jacobian matrix of the generator $G_\theta$. If the step size is chosen appropriately, we expect that the value of $F(z)$ decreases with the iterations. Conceptually, our refinement approach is similar to many second-order optimization routines, which also require a good initialization for convergence.

### 3.4 Regularization for Noisy Data

If the number of training points $M$ is small or if the noise on the $y_k \approx f(x_k)$ is strong, minimizing the empirical risk (3) can suffer from overfitting (i.e., the usage of high-frequency features). To prevent this, we deploy a regularizer of the form (4). Choosing $p = q = 1$ and $D = \nabla$ in (4) leads to the following training problem with (anisotropic) total variation regularization (Acar & Vogel, 1994; Chan & Esedoglu, 2005)

$$\hat{w} \in \arg\min_w \sum_{k=1}^M |y_k - f_{w,b(w)}(x_k)|^2 + \lambda R(w), \quad R(w) \coloneqq \int_{[a_{\min}, a_{\max}]} \|\nabla f_{w,b(w)}(x)\|_1 \mathrm{d}x, \tag{10}$$

where $\lambda > 0$, and $a_{\min} = \min\{x_k : k = 1, ..., M\}$ and $a_{\max} = \max\{x_k : k = 1, ..., M\}$ are the entry-wise minimum and maximum of the training data. For our generative training loss (8), adding the regularizer from (10) leads to

$$\mathcal{L}_{\mathrm{reg}}(\theta) = \mathbb{E}_{w \sim (G_{\theta\#}\eta)^{\otimes N}} [L(w) + \lambda R(w)]. \tag{11}$$

Similarly, we replace the $F$ from (9) for the feature refinement in the latent space by

$$F_{\mathrm{reg}}(z) = F(z) + \lambda R(G_\theta(z)). \tag{12}$$

If we have specific knowledge about the function $f$ that we intend to approximate, then we can apply more restrictive regularizers of the form (10). As discussed in Section 2, several RFMs instead regularize the feature selection by enforcing that the features $w_l \in \mathbb{R}^d$ only have a few non-zero entries (sparse features).

**Remark 1.** *Given the nature of our numerical examples, we only discussed $f \colon [0,1]^d \to \mathbb{R}$ with data points $y_k \in \mathbb{R}$. The extension of our method to multivariate $f \colon [0,1]^d \to \mathbb{R}^n$ is straight forward.*

## 4 Experiments

We demonstrate the effectiveness of our method with three numerical examples. First, we visually inspect the obtained features. Here, we also check if they recover the correct subspaces. Secondly, we benchmark our methods on common test functions from approximation theory, i.e., with a known groundtruth. Lastly, we target regression on some datasets from the UCI database (Kelly et al., 2023).

### 4.1 Setup and Comparisons

For all our experiments, we set up the architecture $f_w, b$ in (1) with $N = 100$ features $(w_l)_{l=1}^N$ and one of the nonlinearities $\Phi$ introduced in Section 3:

- We deploy $\Phi(x) = e^{2\pi i x}$ without the bias trick. This corresponds to the approximation of the underlying ground truth function by Fourier features.

- We deploy $\Phi(x) = \frac{e^x}{1+e^x}$, which corresponds to a 2-layer network with sigmoid activation functions. To improve the expressiveness of the model, we apply the bias trick for both layers.

An ablation for different choices of $N$ is given in Appendix A. Further, we choose the generator $G_\theta$ for the proposal distribution $p_w = G_{\theta\#}\mathcal{N}(0, I_d)$ as ReLU network with 3 hidden layers and 512 neurons per hidden layer. To pick the regularization strength $\lambda$, we divide the original training data into a training (90%) and a validation (10%) set. Then, we train $G_\theta$ for each $\lambda \in \{0\} \cup \{1 \times 10^k : k = -4, ..., 0\}$ and choose the $\lambda$ with the best validation error. To minimize the regularized loss functions $\mathcal{L}_{\text{reg}}$ (GFT, see also (11)) and $F_{\text{reg}}$ (GFT-r, see also (12)), we run 40000 steps of the Adam optimizer. The remaining hyperparameters are given in Appendix C. We benchmark all our methods from Section 3.

- **F-Opt**: In the feature optimization, we minimize the $L(w)$ from (7) with a gradient-based optimizer starting with features $w$ drawn from a standard normal distribution. As explained in Section 3.1, we expect that the optimization gets stuck in a local minimum. We verify this claim in our experiments.

- **GFT**: For the generative feature training as proposed in Section 3.2, we minimize the loss $\mathcal{L}(\theta)$ from (8) and draw iid features $w$ from the generator $G_\theta$ during evaluation.

- **GFT-r**: For the refined generative feature training, we generate features using GFT and refine them with the procedure from Section 3.3. This requires to minimize the loss $F(z)$ in (9).

For each method, we specify the choice of $\Phi$ as "Fourier" and "sigmoid" activation in the corresponding tables. We compare the obtained results with algorithms from the random Fourier feature literature, and with standard training of neural networks. More precisely, we consider the following comparisons:

- **Sparse Fourier Features**: We compare with the random Fourier feature based methods SHRIMP (Xie et al., 2022), HARFE (Saha et al., 2023), SALSA (Kandasamy & Yu, 2016) and ANOVA-boosted random Fourier features (ANOVA-RFF; Potts & Weidensager, 2025). We do not rerun the methods and take the results reported by Xie et al. (2022); Potts & Weidensager (2025).

- **2-Layer Neural Networks**: We train the parameters of the 2-layer neural networks $f_w, b$ with the Adam optimizer. Here, we use exactly the same architecture, loss function and activation function as for GFT. Additionally, we include results for the ReLU activation function $\Phi(x) = \max(x, 0)$.

- **Kernel Ridge Regression**: We perform a kernel ridge regression (Cristianini & Shawe-Taylor, 2000) with the Gaussian kernel, where the kernel parameter is chosen by the median rule.

Our PyTorch implementation is available online[1]. We run all experiments on a NVIDIA RTX 4090 GPU. Depending on the specific model, the training takes between 30 seconds and 2 minutes. We include further ablations in Appendix B.

### 4.2 Visualization of Generated Features

First, we inspect the learned features $w$ in a simple setting. To this end, we consider the function $g: \mathbb{R}^2 \to \mathbb{R}$ with $g(x) = \sin(4\pi x_1^2 + 1) + \cos(4\pi(x_2^4 + x_2))$. Since each summand of $g$ depends either on $x_1$ or $x_2$, its Fourier

---

[1]available at `https://github.com/johertrich/generative_feature_training`

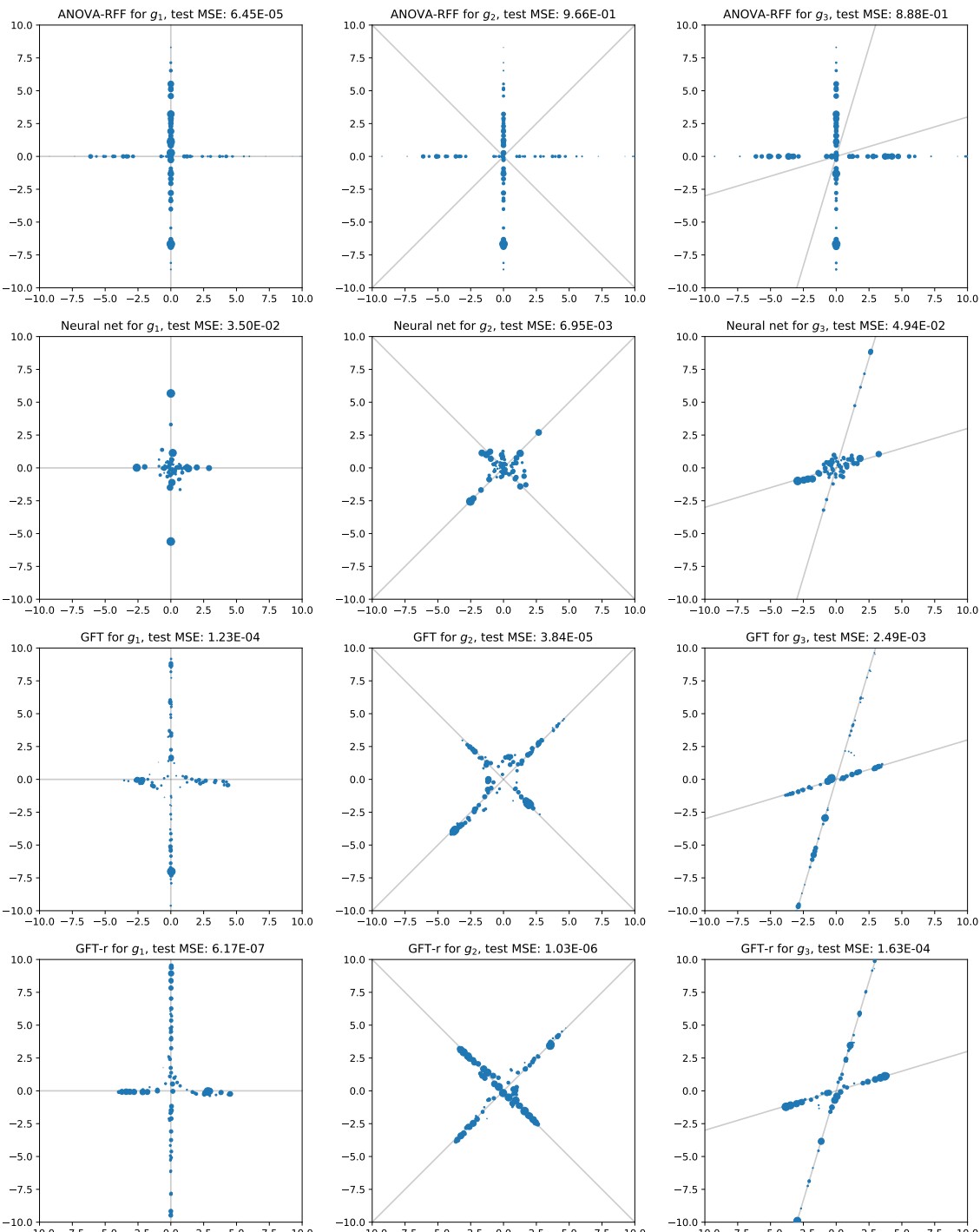

Figure 1: Location of the generated features for the $g_i$ from (13), where the marker size reflects the magnitude of the associated weights $b_l$. The gray lines indicate the support of the Fourier transform of $g_i$. Since ANOVA-RFF constrains the features to be on the axes, it only works for $g_1$. For the standard neural network training, the features are not pushed to the axis. This indicates that the optimization got stuck in a local minimum.

transform is supported on the coordinate axes. To make the task more challenging, we slightly adapt the problem by concatenating $g$ with two linear transforms $A_i$, which leads to the three test functions

$$g_i(x) = g(A_i x), \quad \text{with} \quad A_1 = \begin{pmatrix} 1 & 0 \\ 0 & 1 \end{pmatrix}, \quad A_2 = \begin{pmatrix} \cos(\frac{\pi}{4}) & -\sin(\frac{\pi}{4}) \\ \sin(\frac{\pi}{4}) & \cos(\frac{\pi}{4}) \end{pmatrix}, \quad A_3 = \begin{pmatrix} 1 & 0.3 \\ 0.3 & 1 \end{pmatrix}. \quad (13)$$

In all cases, the Fourier transform is supported on a union of two subspaces. Now, we learn the features $w$ with our GFT and GFT-r method based on 2000 samples that are drawn uniformly from $[0,1]^2$, and plot them in Figure 1. The gray lines indicate the support of the Fourier transforms of $g_i$, and the size of the markers indicates the magnitude of the associated $b_l$. For all functions $g_i$, the features $w$ sampled with GFT are mostly located in the support of the Fourier transform. The very few features that are located outside of it can be explained by numerical errors and are mostly removed by the refinement procedure GFT-r. In contrast, for ANOVA-RFF, the $w$ are restricted to be located on the axes. Consequently, it cannot work for $g_2$ and $g_3$, and the error obtained is large. For gradient-based neural network training, the $w$ are not pushed to the axis, indicating that the optimization got stuck at a local minimum. For functions where the subspaces are orthogonal, such as $g_2$, this issue was recently addressed in Ba et al. (2024) by learning the associated transform in the feature space.

### 4.3 Function Approximation

We use the same experimental setup as in (Potts & Weidensager, 2025, Table 7.1), that is, the test functions

- Polynomial: $f_1(x) = x_4^2 + x_2 x_3 + x_1 x_2 + x_4$;

- Isigami: $f_2(x) = \sin(x_1) + 7 \sin^2(x_2) + 0.1 x_3^4 \sin(x_1)$;

- Friedmann-1: $f_3(x) = 10 \sin(\pi x_1 x_2) + 20(x_3 - \frac{1}{2})^2 + 10 x_4 + 5 x_5$.

The input dimension $d$ is set to 5 or 10 for each $f_k$. In particular, the $f_k$ might not depend on all entries of the input $x$. For their approximation, we are given samples $x_k \sim \mathcal{U}_{[0,1]^d}$, $k = 1, ..., M$, and the corresponding noise-less function values $f_k(x_k)$. The number of samples $M$ and the dimension $d$ are specified for each setting. As test set we draw $M$ additional samples from $\mathcal{U}_{[0,1]^d}$. We deploy our methods as well as standard neural network training to the architecture $f_{w,b}$. The MSEs on the test set are given in Table 1. There, we also include ANOVA-random Fourier features, SHRIMP and HARFE for comparison. We always report the MSE for the best choice of $\rho$ from Potts & Weidensager, 2025, Table 7.1. The GFT-r with Fourier activation functions outperforms the other approaches significantly. In particular, both the GFT and GFT-r consistently improve over the gradient-based training of the architecture $f_{w,b}$. This is in line with the analysis of gradient-based training in recent works (Boob et al., 2022; Holzmüller & Steinwart, 2022). As expected, Fourier activation functions are best suited for this task.

So far, we considered functions $f_i$ that can be represented as sums, where each summand only depends on a small number of inputs $x_i$. While this assumption is crucial for the sparse Fourier feature methods from Table 1, it is not required for our methods. Therefore, we also benchmark them on the following non-decomposable functions and compare the results with standard gradient-based neural network training:

- $h_1(x) = \sin(\sum_{i=1}^d x_i) + \|x\|_2^2$

- $h_2(x) = \sqrt{\|x - \frac{1}{2}e\|_1}$, where $e$ is the vector with all entries equal to one

- $h_3(x) = \sqrt{f_3(x)} = \sqrt{10 \sin(\pi x_1 x_2) + 20(x_3 - \frac{1}{2})^2 + 10 x_4 + 5 x_5}$.

The results are given in Table 2. As in the previous case, we can see a clear advantage of GFT and GFT-r.

Table 1: Comparison with sparse feature methods for function approximation: We report the MSE over the test set averaged over 5 runs. The values for ANOVA-RFF, SHRIMP and HARFE are taken from Potts & Weidensager (2025). The deployed $\lambda$ is indicated below each result. The best performance is highlighted.

| Method | Activation | Function $f_1$ $(d,M)=(5,300)$ | $(d,M)=(10,500)$ | Function $f_2$ $(d,M)=(5,500)$ | $(d,M)=(10,1000)$ | Function $f_3$ $(d,M)=(5,500)$ | $(d,M)=(10,200)$ |
|---|---|---|---|---|---|---|---|
| ANOVA-RFF | Fourier | $1.40 \times 10^{-6}$ | $1.46 \times 10^{-6}$ | $2.65 \times 10^{-5}$ | $2.62 \times 10^{-5}$ | $1.00 \times 10^{-4}$ | $9.80 \times 10^{-3}$ |
| SHRIMP | Fourier | $1.83 \times 10^{-6}$ | $5.00 \times 10^{-4}$ | $8.20 \times 10^{-3}$ | $5.50 \times 10^{-3}$ | $2.00 \times 10^{-4}$ | $3.81 \times 10^{-1}$ |
| HARFE | Fourier | $5.82 \times 10^{-1}$ | $2.38 \times 10^{0}$ | $1.38 \times 10^{-1}$ | $6.65 \times 10^{-1}$ | $3.64 \times 10^{0}$ | $3.98 \times 10^{0}$ |
| kernel ridge reg | | $5.90 \times 10^{-5}$ | $4.40 \times 10^{-4}$ | $7.1 \times 10^{-5}$ | $5.10 \times 10^{-4}$ | $1.15 \times 10^{-2}$ | $1.69 \times 10^{0}$ |
| neural net | Fourier | $2.36 \times 10^{-4}$ ($\lambda=0$) | $1.03 \times 10^{-3}$ ($\lambda=1 \times 10^{-4}$) | $5.28 \times 10^{-5}$ ($\lambda=0$) | $2.23 \times 10^{-4}$ ($\lambda=1 \times 10^{-4}$) | $3.14 \times 10^{-3}$ ($\lambda=1 \times 10^{-4}$) | $2.96 \times 10^{0}$ ($\lambda=1 \times 10^{-4}$) |
| | sigmoid | $3.84 \times 10^{-5}$ ($\lambda=0$) | $5.34 \times 10^{-5}$ ($\lambda=1 \times 10^{-4}$) | $2.25 \times 10^{-5}$ ($\lambda=1 \times 10^{-4}$) | $3.71 \times 10^{-5}$ ($\lambda=1 \times 10^{-4}$) | $2.56 \times 10^{-3}$ ($\lambda=0$) | $2.15 \times 10^{0}$ ($\lambda=1 \times 10^{-3}$) |
| | ReLU | $4.57 \times 10^{-4}$ ($\lambda=1 \times 10^{-4}$) | $1.25 \times 10^{-3}$ ($\lambda=0$) | $1.21 \times 10^{-4}$ ($\lambda=0$) | $1.65 \times 10^{-4}$ ($\lambda=0$) | $6.77 \times 10^{-2}$ ($\lambda=1 \times 10^{-4}$) | $1.55 \times 10^{0}$ ($\lambda=1 \times 10^{-3}$) |
| F-Opt | Fourier | $2.50 \times 10^{-3}$ ($\lambda=1 \times 10^{-4}$) | $1.08 \times 10^{0}$ ($\lambda=1 \times 10^{-3}$) | $2.36 \times 10^{-6}$ ($\lambda=1 \times 10^{-4}$) | $1.31 \times 10^{0}$ ($\lambda=1 \times 10^{-1}$) | $5.93 \times 10^{-2}$ ($\lambda=0$) | $1.68 \times 10^{+1}$ ($\lambda=1 \times 10^{-4}$) |
| | sigmoid | $1.15 \times 10^{-6}$ ($\lambda=0$) | $2.77 \times 10^{-4}$ ($\lambda=0$) | $1.43 \times 10^{-6}$ ($\lambda=0$) | $9.54 \times 10^{-6}$ ($\lambda=0$) | $5.45 \times 10^{-4}$ ($\lambda=0$) | $2.91 \times 10^{0}$ ($\lambda=0$) |
| GFT | Fourier | $2.72 \times 10^{-7}$ ($\lambda=0$) | $5.00 \times 10^{-7}$ ($\lambda=0$) | $1.03 \times 10^{-7}$ ($\lambda=0$) | $4.09 \times 10^{-7}$ ($\lambda=0$) | $5.87 \times 10^{-5}$ ($\lambda=1 \times 10^{-4}$) | $4.47 \times 10^{-3}$ ($\lambda=1 \times 10^{-4}$) |
| | sigmoid | $3.18 \times 10^{-6}$ ($\lambda=0$) | $1.81 \times 10^{-6}$ ($\lambda=0$) | $4.09 \times 10^{-7}$ ($\lambda=0$) | $6.01 \times 10^{-7}$ ($\lambda=0$) | $6.40 \times 10^{-4}$ ($\lambda=0$) | $1.18 \times 10^{-2}$ ($\lambda=1 \times 10^{-4}$) |
| GFT-r | Fourier | $\mathbf{6.05 \times 10^{-8}}$ ($\lambda=0$) | $\mathbf{5.46 \times 10^{-8}}$ ($\lambda=0$) | $\mathbf{2.02 \times 10^{-8}}$ ($\lambda=0$) | $\mathbf{8.15 \times 10^{-8}}$ ($\lambda=0$) | $\mathbf{6.26 \times 10^{-6}}$ ($\lambda=0$) | $\mathbf{1.89 \times 10^{-4}}$ ($\lambda=0$) |
| | sigmoid | $1.05 \times 10^{-6}$ ($\lambda=0$) | $5.60 \times 10^{-7}$ ($\lambda=0$) | $4.97 \times 10^{-8}$ ($\lambda=0$) | $1.12 \times 10^{-7}$ ($\lambda=0$) | $1.50 \times 10^{-5}$ ($\lambda=0$) | $9.94 \times 10^{-3}$ ($\lambda=1 \times 10^{-4}$) |

Table 2: Function approximation: We report the MSE over the test set averaged over 5 runs. The deployed $\lambda$ is indicated below each result. The best performance is highlighted.

| Method | Activation | Function $h_1$ $(d,M)=(10,1000)$ | Function $h_2$ $(d,M)=(20,1000)$ | Function $h_3$ $(d,M)=(5,500)$ |
|---|---|---|---|---|
| kernel ridge reg | | $8.91 \times 10^{-2}$ | $3.74 \times 10^{-3}$ | $5.55 \times 10^{-3}$ |
| neural net | Fourier | $6.03 \times 10^{-2}$ ($\lambda=1 \times 10^{-3}$) | $1.34 \times 10^{-2}$ ($\lambda=0$) | $2.68 \times 10^{-4}$ ($\lambda=0$) |
| | sigmoid | $4.17 \times 10^{-2}$ ($\lambda=0$) | $5.94 \times 10^{-3}$ ($\lambda=1 \times 10^{-4}$) | $4.42 \times 10^{-4}$ ($\lambda=1 \times 10^{-4}$) |
| | ReLU | $5.64 \times 10^{-1}$ ($\lambda=1 \times 10^{-4}$) | $6.89 \times 10^{-3}$ ($\lambda=1 \times 10^{-4}$) | $5.56 \times 10^{-3}$ ($\lambda=1 \times 10^{-3}$) |
| F-Opt | Fourier | $4.27 \times 10^{+1}$ ($\lambda=1 \times 10^{0}$) | $5.06 \times 10^{0}$ ($\lambda=1 \times 10^{-4}$) | $7.24 \times 10^{-3}$ ($\lambda=1 \times 10^{-3}$) |
| | sigmoid | $6.43 \times 10^{-2}$ ($\lambda=0$) | $7.08 \times 10^{-3}$ ($\lambda=0$) | $2.35 \times 10^{-4}$ ($\lambda=0$) |
| GFT | Fourier | $2.62 \times 10^{-2}$ ($\lambda=1 \times 10^{-3}$) | $3.54 \times 10^{-3}$ ($\lambda=1 \times 10^{-4}$) | $2.34 \times 10^{-4}$ ($\lambda=1 \times 10^{-4}$) |
| | sigmoid | $9.36 \times 10^{-2}$ ($\lambda=1 \times 10^{-4}$) | $1.10 \times 10^{-2}$ ($\lambda=1 \times 10^{-4}$) | $4.70 \times 10^{-4}$ ($\lambda=1 \times 10^{-4}$) |
| GFT-r | Fourier | $\mathbf{8.96 \times 10^{-3}}$ ($\lambda=0$) | $\mathbf{2.57 \times 10^{-3}}$ ($\lambda=1 \times 10^{-4}$) | $\mathbf{1.04 \times 10^{-4}}$ ($\lambda=0$) |
| | sigmoid | $6.06 \times 10^{-2}$ ($\lambda=1 \times 10^{-3}$) | $1.00 \times 10^{-2}$ ($\lambda=1 \times 10^{-4}$) | $2.84 \times 10^{-4}$ ($\lambda=1 \times 10^{-4}$) |

## 4.4 Regression on UCI Datasets

Next, we apply our method for regression on several UCI datasets Kelly et al. (2023). For this, we do not have an underlying ground truth function $f$. Here, we compare our methods with standard gradient-based neural network training, SHRIMP and SALSA. To this end, we use the numerical setup of SHRIMP. For each dataset, the MSE on the test split is given in Table 3. Compared to the other methods, SHRIMP and SALSA appear a bit more robust to noise and outliers, which frequently occur in the UCI datasets. This behavior is not surprising, since the enforced sparsity of the features $w_l$ for those methods is a strong

Table 3: Regression on UCI datasets: We report the MSE on the test datasets averaged over 5 runs. The values for SHRIMP and SALSA are taken from Xie et al. (2022). The deployed $\lambda$ is indicated below each result. The best performance is highlighted.

| Method | | Dataset | | | | | |
|---|---|---|---|---|---|---|---|
| Method | Activation | Propulsion $(d,M)=(15,200)$ | Galaxy $(d,M)=(20,2000)$ | Airfoil $(d,M)=(41,750)$ | CCPP $(d,M)=(59,2000)$ | Telemonit $(d,M)=(19,1000)$ | Skillkraft $(d,M)=(18,1700)$ |
| SHRIMP | Fourier | $1.02 \times 10^{-6}$ | $5.41 \times 10^{-6}$ | $2.65 \times 10^{-1}$ | $\mathbf{6.55 \times 10^{-2}}$ | $6.00 \times 10^{-2}$ | $5.81 \times 10^{-1}$ |
| SALSA | Fourier | $8.81 \times 10^{-3}$ | $1.35 \times 10^{-4}$ | $5.18 \times 10^{-1}$ | $6.78 \times 10^{-2}$ | $3.47 \times 10^{-2}$ | $\mathbf{5.47 \times 10^{-1}}$ |
| kernel ridge reg | | $8.60 \times 10^{-3}$ | $2.38 \times 10^{-3}$ | $8.10 \times 10^{-1}$ | $1.24 \times 10^{-1}$ | $1.06 \times 10^{-1}$ | $6.30 \times 10^{0}$ |
| neural net | Fourier | $9.07 \times 10^{-3}$ $(\lambda = 1 \times 10^{-2})$ | $4.46 \times 10^{-4}$ $(\lambda = 1 \times 10^{-4})$ | $3.41 \times 10^{-1}$ $(\lambda = 1 \times 10^{-1})$ | $6.97 \times 10^{-2}$ $(\lambda = 1 \times 10^{-1})$ | $2.51 \times 10^{-2}$ $(\lambda = 1 \times 10^{-3})$ | $6.01 \times 10^{-1}$ $(\lambda = 1 \times 10^{-1})$ |
| | sigmoid | $9.21 \times 10^{-3}$ $(\lambda = 0)$ | $1.67 \times 10^{-4}$ $(\lambda = 1 \times 10^{-4})$ | $3.31 \times 10^{-1}$ $(\lambda = 1 \times 10^{-1})$ | $8.01 \times 10^{-2}$ $(\lambda = 1 \times 10^{-1})$ | $7.86 \times 10^{-2}$ $(\lambda = 1 \times 10^{-3})$ | $1.57 \times 10^{0}$ $(\lambda = 1 \times 10^{-3})$ |
| | ReLU | $5.92 \times 10^{-4}$ $(\lambda = 1 \times 10^{-3})$ | $4.72 \times 10^{-4}$ $(\lambda = 0)$ | $3.66 \times 10^{-1}$ $(\lambda = 1 \times 10^{-1})$ | $6.73 \times 10^{-2}$ $(\lambda = 1 \times 10^{-1})$ | $2.71 \times 10^{-2}$ $(\lambda = 1 \times 10^{-2})$ | $2.23 \times 10^{0}$ $(\lambda = 1 \times 10^{0})$ |
| F-Opt | Fourier | $6.96 \times 10^{-1}$ $(\lambda = 1 \times 10^{-3})$ | $3.51 \times 10^{0}$ $(\lambda = 1 \times 10^{-1})$ | $1.05 \times 10^{0}$ $(\lambda = 1 \times 10^{-4})$ | $9.97 \times 10^{-1}$ $(\lambda = 1 \times 10^{-1})$ | $1.01 \times 10^{0}$ $(\lambda = 1 \times 10^{-1})$ | $1.01 \times 10^{0}$ $(\lambda = 1 \times 10^{0})$ |
| | sigmoid | $1.57 \times 10^{-2}$ $(\lambda = 1 \times 10^{-4})$ | $1.91 \times 10^{-4}$ $(\lambda = 0)$ | $5.92 \times 10^{-1}$ $(\lambda = 1 \times 10^{-3})$ | $7.35 \times 10^{-2}$ $(\lambda = 1 \times 10^{-1})$ | $7.38 \times 10^{-2}$ $(\lambda = 1 \times 10^{-3})$ | $5.79 \times 10^{-1}$ $(\lambda = 1 \times 10^{0})$ |
| GFT | Fourier | $8.31 \times 10^{-7}$ $(\lambda = 0)$ | $3.31 \times 10^{-5}$ $(\lambda = 0)$ | $\mathbf{2.34 \times 10^{-1}}$ $(\lambda = 1 \times 10^{-1})$ | $8.06 \times 10^{-2}$ $(\lambda = 1 \times 10^{-2})$ | $1.05 \times 10^{-2}$ $(\lambda = 1 \times 10^{-2})$ | $5.66 \times 10^{-1}$ $(\lambda = 1 \times 10^{0})$ |
| | sigmoid | $1.22 \times 10^{-5}$ $(\lambda = 0)$ | $7.42 \times 10^{-5}$ $(\lambda = 0)$ | $2.90 \times 10^{-1}$ $(\lambda = 1 \times 10^{-1})$ | $6.86 \times 10^{-2}$ $(\lambda = 1 \times 10^{-1})$ | $1.35 \times 10^{-2}$ $(\lambda = 1 \times 10^{-4})$ | $9.68 \times 10^{-1}$ $(\lambda = 1 \times 10^{-1})$ |
| GFT-r | Fourier | $\mathbf{6.97 \times 10^{-7}}$ $(\lambda = 0)$ | $\mathbf{5.36 \times 10^{-6}}$ $(\lambda = 0)$ | $\mathbf{2.34 \times 10^{-1}}$ $(\lambda = 1 \times 10^{-1})$ | $8.04 \times 10^{-2}$ $(\lambda = 1 \times 10^{-2})$ | $\mathbf{6.48 \times 10^{-3}}$ $(\lambda = 0 \times 10^{0})$ | $5.65 \times 10^{-1}$ $(\lambda = 1 \times 10^{0})$ |
| | sigmoid | $1.67 \times 10^{-5}$ $(\lambda = 0)$ | $1.85 \times 10^{-5}$ $(\lambda = 0)$ | $2.89 \times 10^{-1}$ $(\lambda = 1 \times 10^{-1})$ | $6.84 \times 10^{-2}$ $(\lambda = 1 \times 10^{-1})$ | $9.39 \times 10^{-3}$ $(\lambda = 0)$ | $9.88 \times 10^{-1}$ $(\lambda = 1 \times 10^{-1})$ |

implicit regularization. Incorporating similar sparsity constraints into our generative training is left for future research. Even without such a regularization, GFT-r achieves the best performance on most datasets. Again, both GFT and GFT-r achieve significantly better results than standard training with the Adam optimizer.

# 5 Discussion

**Summary**  We proposed a training procedure for $f_{w,b}$ as in (1) with only a few hidden neurons $w$. In our procedure, we sample the $w$ from a generative model and compute the optimal $b$ by solving a linear system. To enhance the results, we apply a feature refinement scheme in the latent space of the generative model and regularize the loss function. Numerical examples have shown that the proposed generative feature training significantly outperforms standard training procedures.

**Outlook**  Our approach can be extended in several directions. First, we want to train deeper networks in a greedy way similar to (Belilovsky et al., 2019). Recently, a similar approach was considered in the context of sampled networks by Bolager et al. (2023). Moreover, we can encode a sparse structure on the features by replacing the latent distribution $N(0, I_d)$ with a lower-dimensional latent model or by considering mixtures of generative models. From a theoretical side, we want to characterize the global minimizers of the functional in (8) and their relations to the Fourier transform of the target function.

**Limitations**  If $N$ in (1) gets large, solving the linear system (6) becomes expensive. However, this corresponds to the overparameterized regime where gradient-based methods work well. Moreover, the computation of the optimal $b$ depends on all data points. Consequently, if we do minibatching, the output weights $b$ are batch-dependent, and both the theoretical and practical implications remain open. Instead, we emphasize that one motivation for our method is the treatment of small data sets, where no minibatching is required. This is actually also one of the main use cases for 2-layer neural networks. Finally, note that GFT is currently restricted to the $L_2$-loss function, which limits the applicability of GFT to non-regression tasks. For other loss functions, bilevel learning methods could be used to compute and differentiate the optimal output layer. This is beyond the scope of this paper, and we leave this point for future work.

**Acknowledgments**

We would like to thank Daniel Potts, Gabriele Steidl and Laura Weidensager for fruitful discussions. JH acknwoledges funding by the German Research Foundation (DFG) within the Walter Benjamin Programme with project number 530824055 and by the EPSRC programme grant "The Mathematics of Deep Learning" with reference EP/V026259/1.

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

Table 4: Comparison for function approximation with: We report the MSE over the test set averaged over 5 runs. The table contains the same experiments as Table 1 with $N = 50$ features. The deployed $\lambda$ is indicated below each result. The best performance is highlighted.

| Method | Activation | Function $f_1$ | | Function $f_2$ | | Function $f_3$ | |
|---|---|---|---|---|---|---|---|
| | | $(d, M) = (5, 300)$ | $(d, M) = (10, 500)$ | $(d, M) = (5, 500)$ | $(d, M) = (10, 1000)$ | $(d, M) = (5, 500)$ | $(d, M) = (10, 200)$ |
| neural net | Fourier | $3.35 \times 10^{-5}$ ($\lambda = 0$) | $9.22 \times 10^{-5}$ ($\lambda = 1 \times 10^{-4}$) | $6.97 \times 10^{-6}$ ($\lambda = 0$) | $8.15 \times 10^{-6}$ ($\lambda = 0$) | $2.45 \times 10^{-3}$ ($\lambda = 1 \times 10^{-4}$) | $4.65 \times 10^{0}$ ($\lambda = 1 \times 10^{-3}$) |
| | sigmoid | $1.03 \times 10^{-5}$ ($\lambda = 0$) | $1.52 \times 10^{-5}$ ($\lambda = 0$) | $1.19 \times 10^{-5}$ ($\lambda = 0$) | $1.45 \times 10^{-5}$ ($\lambda = 1 \times 10^{-4}$) | $\mathbf{2.14 \times 10^{-3}}$ ($\lambda = 0$) | $2.27 \times 10^{0}$ ($\lambda = 1 \times 10^{-4}$) |
| | ReLU | $4.71 \times 10^{-4}$ ($\lambda = 0$) | $1.40 \times 10^{-3}$ ($\lambda = 0$) | $3.00 \times 10^{-4}$ ($\lambda = 0$) | $1.12 \times 10^{-4}$ ($\lambda = 1 \times 10^{-4}$) | $2.08 \times 10^{-1}$ ($\lambda = 0$) | $1.89 \times 10^{0}$ ($\lambda = 1 \times 10^{-4}$) |
| F-Opt | Fourier | $1.60 \times 10^{-5}$ ($\lambda = 1 \times 10^{-4}$) | $2.52 \times 10^{0}$ ($\lambda = 0$) | $5.01 \times 10^{-6}$ ($\lambda = 0$) | $7.45 \times 10^{0}$ ($\lambda = 0$) | $5.13 \times 10^{-3}$ ($\lambda = 1 \times 10^{-4}$) | $2.60 \times 10^{+2}$ ($\lambda = 1 \times 10^{-4}$) |
| | sigmoid | $2.44 \times 10^{-4}$ ($\lambda = 0$) | $8.05 \times 10^{-5}$ ($\lambda = 1 \times 10^{-4}$) | $3.58 \times 10^{-3}$ ($\lambda = 1 \times 10^{-4}$) | $1.69 \times 10^{-6}$ ($\lambda = 0$) | $1.08 \times 10^{0}$ ($\lambda = 1 \times 10^{-4}$) | $3.94 \times 10^{0}$ ($\lambda = 1 \times 10^{-4}$) |
| GFT | Fourier | $9.49 \times 10^{-5}$ ($\lambda = 0$) | $4.19 \times 10^{-5}$ ($\lambda = 1 \times 10^{-3}$) | $5.71 \times 10^{-5}$ ($\lambda = 1 \times 10^{-3}$) | $1.98 \times 10^{-5}$ ($\lambda = 1 \times 10^{-4}$) | $9.17 \times 10^{-2}$ ($\lambda = 1 \times 10^{-3}$) | $8.72 \times 10^{-3}$ ($\lambda = 1 \times 10^{-4}$) |
| | sigmoid | $1.94 \times 10^{-1}$ ($\lambda = 1 \times 10^{-4}$) | $6.08 \times 10^{-5}$ ($\lambda = 1 \times 10^{-3}$) | $5.41 \times 10^{-1}$ ($\lambda = 1 \times 10^{-4}$) | $9.01 \times 10^{-4}$ ($\lambda = 1 \times 10^{-2}$) | $9.81 \times 10^{0}$ ($\lambda = 1 \times 10^{-4}$) | $1.09 \times 10^{-2}$ ($\lambda = 0$) |
| GFT-r | Fourier | $\mathbf{5.52 \times 10^{-6}}$ ($\lambda = 0$) | $\mathbf{2.94 \times 10^{-7}}$ ($\lambda = 0$) | $\mathbf{1.24 \times 10^{-6}}$ ($\lambda = 0$) | $\mathbf{9.75 \times 10^{-7}}$ ($\lambda = 1 \times 10^{-4}$) | $8.72 \times 10^{-3}$ ($\lambda = 0$) | $\mathbf{2.02 \times 10^{-4}}$ ($\lambda = 0$) |
| | sigmoid | $1.94 \times 10^{-1}$ ($\lambda = 1 \times 10^{-4}$) | $6.96 \times 10^{-2}$ ($\lambda = 0$) | $4.33 \times 10^{-1}$ ($\lambda = 0$) | $4.78 \times 10^{-1}$ ($\lambda = 1 \times 10^{-4}$) | $2.66 \times 10^{0}$ ($\lambda = 0$) | $2.82 \times 10^{-3}$ ($\lambda = 0$) |

Daniel Potts and Laura Weidensager. ANOVA-boosting for random Fourier features. *Applied and Computational Harmonic Analysis*, 79:101789, 2025.

Daniel Potts, Gabriele Steidl, and Manfred Tasche. Fast Fourier transforms for nonequispaced data: A tutorial. *Modern Sampling Theory: Mathematics and Applications*, pp. 247–270, 2001.

Ali Rahimi and Benjamin Recht. Random features for large-scale kernel machines. In *Advances in Neural Information Processing Systems*, volume 20, 2007.

Ali Rahimi and Benjamin Recht. Uniform approximation of functions with random bases. In *46th Annual Allerton Conference on Communication, Control, and Computing*, pp. 555–561. IEEE, 2008.

Alessandro Rudi and Lorenzo Rosasco. Generalization properties of learning with random features. In *Advances in Neural Information Processing Systems*, volume 30, 2017.

Nicolaj Rux, Michael Quellmalz, and Gabriele Steidl. Slicing of radial functions: a dimension walk in the Fourier space. *Sampling Theory, Signal Processing, and Data Analysis*, 23(1):1–40, 2025.

Esha Saha, Hayden Schaeffer, and Giang Tran. HARFE: Hard-ridge random feature expansion. *Sampling Theory, Signal Processing, and Data Analysis*, 21(2):27, 2023.

Yuege Xie, Robert Shi, Hayden Schaeffer, and Rachel Ward. SHRIMP: Sparser random feature models via iterative magnitude pruning. In *Proceedings of Mathematical and Scientific Machine Learning*, volume 190, pp. 303–318. PMLR, 2022.

Ian En-Hsu Yen, Ting-Wei Lin, Shou-De Lin, Pradeep K Ravikumar, and Inderjit S Dhillon. Sparse random feature algorithm as coordinate descent in Hilbert space. In *Advances in Neural Information Processing Systems*, volume 27, 2014.

## A Dependence on the Number of Features

We redo the experiments from Section 4.3 for $N = 50$ and $N = 200$. The results are given in Table 4 and 5.

Table 5: Comparison for function approximation with: We report the MSE over the test set averaged over 5 runs. The table contains the same experiments as Table 1 with $N = 200$ features. The deployed $\lambda$ is indicated below each result. The best performance is highlighted.

| Method | | Function $f_1$ | | Function $f_2$ | | Function $f_3$ | |
|---|---|---|---|---|---|---|---|
| Method | Activation | $(d, M) = (5, 300)$ | $(d, M) = (10, 500)$ | $(d, M) = (5, 500)$ | $(d, M) = (10, 1000)$ | $(d, M) = (5, 500)$ | $(d, M) = (10, 200)$ |
| neural net | Fourier | $7.22 \times 10^{-4}$ $(\lambda = 0)$ | $1.21 \times 10^{-2}$ $(\lambda = 1 \times 10^{-4})$ | $1.56 \times 10^{-4}$ $(\lambda = 1 \times 10^{-4})$ | $1.36 \times 10^{-4}$ $(\lambda = 1 \times 10^{-4})$ | $2.26 \times 10^{-3}$ $(\lambda = 1 \times 10^{-4})$ | $3.95 \times 10^{0}$ $(\lambda = 0)$ |
| | sigmoid | $1.54 \times 10^{-5}$ $(\lambda = 0)$ | $2.35 \times 10^{-5}$ $(\lambda = 0)$ | $1.46 \times 10^{-5}$ $(\lambda = 1 \times 10^{-4})$ | $4.91 \times 10^{-5}$ $(\lambda = 0)$ | $1.21 \times 10^{-3}$ $(\lambda = 1 \times 10^{-4})$ | $1.79 \times 10^{0}$ $(\lambda = 0)$ |
| | ReLU | $3.03 \times 10^{-4}$ $(\lambda = 0)$ | $1.05 \times 10^{-3}$ $(\lambda = 0)$ | $1.78 \times 10^{-4}$ $(\lambda = 0)$ | $2.02 \times 10^{-4}$ $(\lambda = 1 \times 10^{-4})$ | $6.20 \times 10^{-2}$ $(\lambda = 1 \times 10^{-4})$ | $1.84 \times 10^{0}$ $(\lambda = 1 \times 10^{-4})$ |
| F-Opt | Fourier | $9.45 \times 10^{-3}$ $(\lambda = 1 \times 10^{-3})$ | $1.52 \times 10^{-2}$ $(\lambda = 1 \times 10^{-2})$ | $4.47 \times 10^{-5}$ $(\lambda = 1 \times 10^{-4})$ | $2.34 \times 10^{-2}$ $(\lambda = 1 \times 10^{-2})$ | $1.29 \times 10^{0}$ $(\lambda = 1 \times 10^{-2})$ | $2.11 \times 10^{+1}$ $(\lambda = 1 \times 10^{0})$ |
| | sigmoid | $2.32 \times 10^{-6}$ $(\lambda = 0)$ | $5.73 \times 10^{-4}$ $(\lambda = 1 \times 10^{-4})$ | $1.19 \times 10^{-6}$ $(\lambda = 0)$ | $8.92 \times 10^{-5}$ $(\lambda = 0)$ | $7.23 \times 10^{-4}$ $(\lambda = 0)$ | $2.40 \times 10^{0}$ $(\lambda = 0)$ |
| GFT | Fourier | $6.66 \times 10^{-7}$ $(\lambda = 0)$ | $2.33 \times 10^{-7}$ $(\lambda = 0)$ | $5.73 \times 10^{-8}$ $(\lambda = 0)$ | $6.53 \times 10^{-7}$ $(\lambda = 0)$ | $2.26 \times 10^{-5}$ $(\lambda = 0)$ | $1.51 \times 10^{0}$ $(\lambda = 1 \times 10^{-4})$ |
| | sigmoid | $2.84 \times 10^{-6}$ $(\lambda = 0)$ | $3.52 \times 10^{-6}$ $(\lambda = 0)$ | $2.04 \times 10^{-7}$ $(\lambda = 0)$ | $3.05 \times 10^{-7}$ $(\lambda = 0)$ | $1.62 \times 10^{-4}$ $(\lambda = 0)$ | $1.75 \times 10^{-2}$ $(\lambda = 0)$ |
| GFT-r | Fourier | $\mathbf{1.57 \times 10^{-7}}$ $(\lambda = 0)$ | $\mathbf{6.01 \times 10^{-8}}$ $(\lambda = 0)$ | $\mathbf{1.42 \times 10^{-8}}$ $(\lambda = 0)$ | $\mathbf{1.66 \times 10^{-7}}$ $(\lambda = 0)$ | $\mathbf{1.28 \times 10^{-6}}$ $(\lambda = 0)$ | $1.82 \times 10^{0}$ $(\lambda = 1 \times 10^{-4})$ |
| | sigmoid | $2.04 \times 10^{-6}$ $(\lambda = 0)$ | $2.26 \times 10^{-6}$ $(\lambda = 0)$ | $4.27 \times 10^{-8}$ $(\lambda = 0)$ | $5.41 \times 10^{-8}$ $(\lambda = 0)$ | $1.84 \times 10^{-5}$ $(\lambda = 0)$ | $\mathbf{1.60 \times 10^{-2}}$ $(\lambda = 0)$ |

Table 6: Comparison for function approximation with: We report the MSE over the test set averaged over 5 runs. The table contains the same experiments as Table 1 for the two ablations in Appendix B. The deployed $\lambda$ is indicated below each result.

| Method | | Function $f_1$ | | Function $f_2$ | | Function $f_3$ | |
|---|---|---|---|---|---|---|---|
| Method | Activation | $(d, M) = (5, 300)$ | $(d, M) = (10, 500)$ | $(d, M) = (5, 500)$ | $(d, M) = (10, 1000)$ | $(d, M) = (5, 500)$ | $(d, M) = (10, 200)$ |
| Fully Sampled | Fourier | $1.41 \times 10^{-3}$ $(\lambda = 1 \times 10^{-4})$ | $1.55 \times 10^{-2}$ $(\lambda = 1 \times 10^{-3})$ | $4.01 \times 10^{-2}$ $(\lambda = 1 \times 10^{-3})$ | $3.73 \times 10^{-2}$ $(\lambda = 1 \times 10^{-4})$ | $4.89 \times 10^{0}$ $(\lambda = 0)$ | $5.70 \times 10^{0}$ $(\lambda = 1 \times 10^{-1})$ |
| Noisy gradients | Fourier | $1.03 \times 10^{-2}$ $(\lambda = 0)$ | $7.56 \times 10^{-3}$ $(\lambda = 1 \times 10^{-2})$ | $7.04 \times 10^{-3}$ $(\lambda = 0)$ | $4.16 \times 10^{-3}$ $(\lambda = 1 \times 10^{-2})$ | $1.38 \times 10^{-2}$ $(\lambda = 1 \times 10^{-4})$ | $7.83 \times 10^{0}$ $(\lambda = 1 \times 10^{-4})$ |

## B  Further Ablations

First, to demonstrate that using the optimal output weights $b(w)$ from Section 3.1 is necessary for GFT, we run the following experiment: Instead of sampling only features $w_l$ from a generative model $p_w = G_{\theta \#} \eta$ with $G_\theta \colon \mathbb{R}^d \to \mathbb{R}^d$ and then computing $b(w)$, we sample pairs $(w_l, b_l)$ from $p_{w,b} = \tilde{G}_{\theta \#} \tilde{\eta}$ for $\tilde{G}_\theta \colon \mathbb{R}^{d+2} \to \mathbb{R}^d \times \mathbb{C}$ and the $d+2$ dimensional standard normal distribution $\tilde{\eta}$. We use the Fourier activation and the same setup as for Table 1. The results in the first row of Table 6 are significantly worse than for GFT and GFT-r in Table 1. This is not surprising since now each output weight $b_l$ only depends on its corresponding feature $w_l$ instead of all $(w_l)_{l=1}^N$ as for the optimal $b(w)$. In particular, $b(w)$ cannot be learned by a gradient-based algorithms since in each update the $w = (w_l)_{l=1}^N$ are resampled. Thus, there is no persisting correspondence of sampled features with specific output weights.

Second, we test whether the neural network line in Table 1 can be improved by introducing gradient noise. This technique can help to escape local minima. To this end, we optimize the same loss function as for the 2-layer neural network in Table 1, but use the noisy stochastic gradient descent $\theta_{n+1} = \theta_n - \rho \nabla L(\theta) + \alpha Z$ with random $Z \sim \mathcal{N}(0, I)$, step size $\rho$, and noise strength $\alpha = 0.1\rho$. The obtained results in the second row of Table 6 do not improve upon the baseline from Table 1.

## C  Implementation Details

We optimize the loss functions for GFT and for the feature refinement with the Adam optimizer using a learning rate of $1 \times 10^{-4}$ for 40000 steps. The regularization $\epsilon$ for solving the least squares problem (6) is

set to $\epsilon = 1 \times 10^{-7}$. For the neural network optimization, we use the Adam optimizer with a learning rate of $1 \times 10^{-3}$ for 100000 steps. In all cases, we discretize the spatial integral for the regularization term in (10) by 1000 samples. For the kernel ridge regression, we use a Gauss kernel with its parameter chosen by the median rule. That is, we set it to the median distance of two points in the dataset. The PyTorch implementation corresponding to our experiments is available at `https://github.com/johertrich/generative_feature_training`.

