# OpenReview forum: "Generative Feature Training of Thin 2-Layer Networks"
_TMLR — Accepted by TMLR_

### Review · Reviewer_1jSb · 2024-12-27

**Summary Of Contributions:**

This work proposes GFT, a novel approach to train narrow and shallow (2-layer) neural networks as function approximators. Specifically, the method first learns a distribution that generates random weights for the hidden layer, parameterized by a deep neural network. Then, it fine-tunes the hidden layer weights. Throughout the process, the output layer weights are optimized in closed form. The authors show that GFT achieves smaller errors when used as function approximators on various datasets compared with existing methods.

**Audience:**

Yes

**Claims And Evidence:**

Yes

**Requested Changes:**

- Please add some explanations on why we should be interested in optimizing two-layer thin neural networks. Given that we train a weight generator $G_\theta$ that is a deep neural network in the process of learning $f_{w, b}$, why don't we just use a deep neural network for our desired task? One scenario I can think of is that $f_{w, b}$ needs to be deployed onto an embedded device with extremely limited system resources.
- The authors mention that "For all other hyperparameters, we refer to our code." I appreciate open-sourcing the code. However, the paper should also be self-contained, so I suggest the authors include the hyperparameters in the appendix.
- Please address the weaknesses and questions above.

**Strengths And Weaknesses:**

Strengths:
- The proposed method is novel and intriguing.

Weaknesses:
- Typos and unclear notations:
  - Page 3: $\in$ became $in$.
  - Page 3: Does "AD" stand for auto-differentiation? This is not explained in the paper.
  - $L$: $L$ denotes a loss function in Eq.(7). However, $L$ seems to denote something else in Eq.(10). Please make sure that notations are consistent.
  - GFT-p: According to the paper, "p" here stands for "post-processing". However, what "post-processing" refers to is not clearly explained. Is it the refinement in Section 3.3?
  - Ground-truth function: Earlier in the paper, the true underlying function we try to approximate used the notation $f$. However, in Section 4.2, the notation became $g$. It should be consistent.
  - Page 10: "we computation".

- Ablation study:
  - All experiments use $N=100$ features. How does GFT perform when $N$ is changed to some other values? How about traditional back-propagation?
  - GFT consists of two stages of the training: $\theta$-optimiation and $w$-finetuning. How much does each stage contribute to the model performance? How does the proposed method perform without the $\theta$-optimization step?

- All experiments consider small statistical learning datasets. One may argue that these smaller tasks are often handled by models based on convex optimization, such as kernel regression. Therefore, I suggest the authors include additional experiments in at least one of the following two directions:
  - On the smaller datasets used in the current manuscript, compare with some statistical learning tools such as kernel regression.
  - Neural networks (even smaller ones) are more commonly used in more complex tasks such as image classification/regression. How does GFT perform on those? Classification tasks include MNIST and CIFAR-10. One possible image regression task to consider is aesthetics score prediction (similar to LAION aesthetics prediction [2]).

Questions:
- Since the hidden weights are fixed except the feature fine-tuning phase, how is GFT related to Extreme Learning Machine (ELM) [1]?
- In each GFT plot in Figure 1, there seem to be some dots in the middle of each plot that are not on either axes, and this only occurs to GFT and GFT-p. Is there any thoughts on why it happens?
- Can GFT be adapted to train deeper neural networks (e.g., three layers)? Can it be used to train non-regression models?
- The authors mention that "the computation of the optimal output layer requires to consider all data points at once, such that we cannot use minibatching". Why is this the case? Wouldn't it be possible to use a random subset of the dataset to perform each optimization step?

[1] Huang et al. Extreme learning machine: a new learning scheme of feedforward neural networks. \
[2] Schuhmann et al. Improved Aesthetic Predictor. https://laion.ai/blog/laion-aesthetics

---

> ### Author Response · Authors · 2025-03-06
> **Response to the Review 1/2**
>
> We would like to thank the reviewer for the thorough evaluation of our paper. Please find below our answers to the raised questions and weaknesses.
>
> > Typos and unclear notations
>
> Many thanks for pointing this out. We corrected the typos. To unify the notation, we renamed GFT-p to GFT-r, since the postprocessing is called refinement before. To make the notation more clear, we rewrote the corresponding part in Section 4.1 and list precisely which abbreviation (GFT, GFT-r and the now added F-Opt) correspond to the minimization of which loss function in Section 3.
>
> > Ground-truth function: Earlier in the paper, the true underlying function we try to approximate used the notation $f$. However, in Section 4.2, the notation became $g$. It should be consistent.
>
> In the numerical section, we consider different types of ground truth functions. The functions $f$ are ANOVA-type functions which can be approximated by sparse Fourier features. To demonstrate the advantage of GFT above sparse feature methods, we also consider functions $g$ where the Fourier transform is still sparse, but no longer aligned with the axes. Finally, the functions $h$ do not have any sparse representation of the Fourier space. We use different letters for the ground truth functions to clearly distinguish these cases.
>
> > Ablation study
>
> Thanks for the suggestion. We added ablations for $N=50$ and $N=200$ features in the appendix. Moreover, we added the vanilla feature refinement ("feature optimization (F-Opt)") from Section 3.3 with random initialization (instead of the $\theta$-optimization).
>
> > All experiments consider small statistical learning datasets. One may argue that these smaller tasks are often handled by models based on convex optimization, such as kernel regression. Therefore, I suggest the authors include additional experiments in at least one of the following two directions:
>
> We added a kernel ridge regression with Gaussian kernel to all tables in the main paper. We observe that our proposed generative feature training with refinement outperforms the kernel ridge regression in all cases.
>
> > Since the hidden weights are fixed except the feature fine-tuning phase, how is GFT related to Extreme Learning Machine (ELM) [1]?
>
> For our method, the hidden weights $w$ are not actually fixed; they are repeatedly sampled from a proposal distribution, where we make use of the explicit formula for the optimal output weights.
> In the refinement phase, we then explicitly optimize starting from a set of sampled weights to get an even better fit.
> In contrast, ELMs are essentially a random feature method, where the weights are only sampled once in the beginning of the training phase.
> Consequently, we feel that ELMs are only loosly related and prefer not to discuss them in the paper.
>
> > In each GFT plot in Figure 1, there seem to be some dots in the middle of each plot that are not on either axes, and this only occurs to GFT and GFT-p. Is there any thoughts on why it happens?
>
> First, we would like to stress that for ANOVA-RFF the features are located by definition on the axes. While this works very well as long as the Fourier transform is supported on these subspaces (see the Figure 1 on the left). Otherwise it leads to very large errors (see Figure 1 middle and right).
> In contrast, GFT(-r) finds the axes numerically such that it can also be used for a much larger function class. As a downside, numerical errors can appear leading to features not located on either of the axes. Two possible reasons for that are the following:
>
> - The center of the feature space corresponds to low frequencies. In particular, the effect of deviations from the axes in this area are comparably small. In particular, in this around the center of the frequency space a slight deviations correspond also only to slight errors.
> - It is topologically difficult for generative models to learn degenerated distributions (which are supported on a zero-set) such that the axis are a bit blurred for GFT.
>
> However, we can clearly see that these numerical errors are mostly resolved by the feature refinement such that for GFT-r almost no features are located outside the subspaces.
>
> We extended the discussion of the example. Moreover, following a comment of Reviewer kQLY, we added also the features arising from the gradient-based training of the neural network to the figure. We can see that it leads to significantly more features outside of either of the axes.

---

> > ### Author Response · Authors · 2025-03-06
> > **Response to the Review 2/2**
> >
> > > Can GFT be adapted to train deeper neural networks (e.g., three layers)? Can it be used to train non-regression models?
> >
> > This is indeed an interesting direction for extending our approach.  We now mention this in the outlook in Section 5. More generally, we could sample all weights despite the output-layer from a generative model. However, for more than one hidden layer, we do not have any longer the nice relation to random feature models. We will consider this idea in the future.
> >
> > > The authors mention that "the computation of the optimal output layer requires to consider all data points at once, such that we cannot use minibatching". Why is this the case? Wouldn't it be possible to use a random subset of the dataset to perform each optimization step?
> >
> > Generally, we could use a random subset of the dataset. However, in this case, the optimal output weights are batch-dependent. From a theoretical side, it is a bit unclear what this leads to. In particular, we are not clear if the arising gradient estimator is still unbiased as it is usually the case for minibatching.
> >
> > We tried it on some of the datasets. Usually, if the corresponding batch size remains large enough, minibatching leads to a slight but not drastic drop of the results. We leave a further investigation of this issue for the future. If the batch size becomes too small, the output weights are no longer meaningful and the method breaks.
> >
> > However, we stress that one main motivation of our method is the treatment of small data sets, where minibatching is not relevant.
> > We rewrote the corresponding part in the Discussion to clarify these points.
> >
> > > Please add some explanations on why we should be interested in optimizing two-layer thin neural networks. Given that we train a weight generator $G_\theta$ that is a deep neural network in the process of learning $f_{w,b}$, why don't we just use a deep neural network for our desired task? One scenario I can think of is that $f_{w,b}$ needs to be deployed onto an embedded device with extremely limited system resources.
> >
> > This would be indeed one particular use case.
> > To provide a more general perspective, we added the following brief discussion to the introduction:
> > Since the Pareto principle suggests that most real-world systems are driven by a few low-complexity interactions, we are interested in representations (1) with only a few features $w_l$.
> > Such an explicit restriction to small $N$ also mitigates overfitting, as seen in sparse neural networks, compressed sensing and feature selection.
> >
> > > The authors mention that "For all other hyperparameters, we refer to our code." I appreciate open-sourcing the code. However, the paper should also be self-contained, so I suggest the authors include the hyperparameters in the appendix.
> >
> > We added a detailed description of all hyperparameters in the appendix.

---

> > > ### Comment · Reviewer_1jSb · 2025-04-19
> > > **Thank you for the response.**
> > >
> > > I appreciate the response from the authors. I have several additional suggestions:
> > >
> > > - Regarding the relationship with ELM, I still think ELM should be discussed. While the authors mention that GFT can fine-tune hidden layer weights post-hoc, the submission indicates that this step is optional. Omitting this step makes GFT related to ELM in that both methods generate hidden layer weights randomly (with the distribution learnable with GFT).
> > > - Regarding Figure 1, I understand that GFT can solve more problems successfully compared with ANOVA-RFF. However, for the first example, where the Fourier transform is supported on axes, I expected GFT to work similarly to ANOVA-RFF. In contrast, Figure 1 shows ANOVA-RFF produces a somewhat cleaner result. I would appreciate some explanations about this observation.
> > > - Can GFT be used to train non-regression models?
> > >
> > > Thank you!

---

> > > > ### Author Response · Authors · 2025-05-03
> > > >
> > > > Thanks for your additional comments. We apologize for the late reply due to traveling and conference trips of the authors.
> > > >
> > > > > Regarding the relationship with ELM
> > > >
> > > > We stress that the main contribution of our paper is to learn the feature distribution. In particular, even without the refinement step, the hidden layer weights are updated during the training by resampling them from the updated feature distribution. Such an update is not considered in ELMs such that we do not see a close connection. Still, we have added a short comment in the related work section.
> > > >
> > > > > Regarding Figure 1
> > > >
> > > > Since we are only given finitely many data points, GFT cannot learn the feature distribution exactly. In particular, this introduces numerical errors such that features are not *exactly* located on the axes.
> > > > In contrast, for ANOVA-RFFs this is inserted as prior information. Consequently, the result of ANOVA-RFFs might look cleaner in this case.
> > > > However, ANOVA-RFF does not adjust the feature positions within these subspaces as it is done for our refinement.
> > > > Therefore, the GFT-r result gives a significantly better MSE than ANOVA-RFFs.
> > > >
> > > > Moreover, one of the main difficulties for ANOVA-RFFs is the a priori identification of the relevant subspaces.
> > > > In particular, this becomes tricky for high dimension as the number of possible subspaces increases rapidly with the dimension.
> > > > For details, we refer to (Potts and Weidensager, 2024), (Potts and Schmischke, 2021) and the references therein.
> > > >
> > > > > Can GFT be used to train non-regression models?
> > > >
> > > > Non-regression tasks are often tackled by some reformulation such that regression models can be applied (e.g. for classification label).
> > > > Once this is done, we can in principle use GFT for non-regression tasks.
> > > > There are two points to be considered:
> > > >
> > > > - For non-regression tasks one often uses other loss functions than the $L^2$ error (e.g. cross entropy for classification). In this case, the computation of the optimal output weights is no longer straight-forward. While it might also be possible to compute them for some other special cases of loss functions, or to employ some deep-equilibrium methods for computing and differentiating the optimal output weights, we think that this is beyond the scope of our paper.
> > > >
> > > > - We often require more than one output dimension. This point can easily be implemented into the generative feature training.
> > > >
> > > > We added a new remark add the end of section 3 stating that also multivariate functions can be approximated by GFT and now mention the restriction to the $L_2$ loss function in the limitations.

---

> ### Comment · Reviewer_1jSb · 2025-05-15
> **Thank you for the clarification**
>
> Thank you for the clarification! My questions and concerns have been largely addressed.
>
> Regarding ELM though, the authors mention that
> > The hidden layer weights are updated during the training by resampling them from the updated feature distribution.
>
> However, the paper (Section 3.2) states that
> >In each step, we sample **one** realization $z \sim \eta \otimes N$ of the latent features to get an estimate for the expectation in (8).
>
> Based on this statement, my understanding is that the distribution that proposes the hidden layer weights is updated during, but the hidden layer weights themselves are not directly optimized. Given each feature distribution, the hidden layer weights are only sampled once, and **NOT resampled** until we change the feature distribution. Is my understanding correct?
>
> If this is correct, I think the proposed method is, in some sense, a "learnable ELM", although I agree that there are multiple different interpretations to this. My apologies in advance if my understanding is incorrect. In any case, I am glad to see that ELM is now mentioned to complete the related work section.

---

> > ### Author Response · Authors · 2025-05-26
> >
> > Thanks for the reply!
> >
> > When we minimize the loss function (8), we update the network $G_\theta$ and therefore the feature distribution in every optimization step. If we kept the samples $z$ from the latent distribution fixed, this corresponds to gradient-based training of the features which is not done in ELM. Since we have the expectation, we resample the latent variables $z$ in each optimization step. In particular, we update the feature distribution based on gradient evaluations (which is not used at all in ELM).
> >
> > Nevertheless, it is true that within each optimization step, we solve the problem (6), which appears in a similar form also for ELM. However, we stress the different scope of this subproblem: The main idea of ELM is that due to the random sampling and easy (solving a linear system) nature of the problem (6), they can take an incredibly large hidden dimension provided that there is enough data available. Here, we are explicitly targeting the case of thin networks, where this problem becomes even cheaper such that we can solve it in each step of a gradient-based algorithm and differentiate its solution.
> >
> > We hope this clarifies the point!

---

### Review · Reviewer_kQLY · 2025-01-07

**Summary Of Contributions:**

The paper proposes an alternative paradigm for function approximation by 2-layer neural networks with small hidden dimension. Concretely, rather than train all weights in the network simultaneously by backpropagation, the proposed paradigm first casts the optimal output weights as a linear function of given fixed hidden weights and learns a proposal distribution for the hidden weights via gradient-based optimization of a generator network (a feedforward a network applied to sampled latents, allowing for reparametrization gradients). Then, the sampled hidden weights $w$ are updated in a refinement step where both the generator network and sampled latents $z$ (treated as parameters) are updated via backpropagation.

**Audience:**

Yes

**Claims And Evidence:**

Yes

**Requested Changes:**

See weaknesses above.

**Strengths And Weaknesses:**

Strengths:
1. The proposed approach, Generative Feature Training (GFT), addresses a fundamental challenge in machine learning (i.e., difficulty of optimizing non-convex landscapes), which could have wide-ranging applications / implications in the field.
2. GFT is, to my knowledge, a novel contribution, dividing the problem into learning an initial proposal distribution and then refining it.
3. The proposed approach generally outperforms existing sparse feature methods for function approximation, including random feature models and neural networks with small hidden dimension trained with backpropagation.

Weaknesses:
1. While the paper does a good job providing a wide range of related work, it would be helpful if the authors could more explicitly situate their contribution / ideas relative to the related work, e.g., problems with existing methods that this method addresses / bypasses, key ideas in other works that are shared by this work. The current drat has this discussion to some extent, but a more systematic discussion would help.
2. It would also be helpful to provide additional motivations and intuitions for various details in the proposed approach. While there is some in the paper, sections 1-3 still seem more procedural than expositional in the current state. Ablations in the experiments would also help in this regard. As another example, figure 1 is quite illustrative in comparing GFT with ANOVA-RFF; and an analogous visual describing how GFT improves over backprop (e.g., potentially exemplar loss curves).

---

> ### Author Response · Authors · 2025-03-06
> **Response to the Review**
>
> Thank you very much for your review. We answer the two points from the weaknesses part separately.
>
> > While the paper does a good job providing a wide range of related work, it would be helpful if the authors could more explicitly situate their contribution / ideas relative to the related work, e.g., problems with existing methods that this method addresses / bypasses, key ideas in other works that are shared by this work. The current drat has this discussion to some extent, but a more systematic discussion would help.
>
> We increased the amount of discussion and also made links with some observations from our paper.
>
>
> > It would also be helpful to provide additional motivations and intuitions for various details in the proposed approach. While there is some in the paper, sections 1-3 still seem more procedural than expositional in the current state. Ablations in the experiments would also help in this regard. As another example, figure 1 is quite illustrative in comparing GFT with ANOVA-RFF; and an analogous visual describing how GFT improves over backprop (e.g., potentially exemplar loss curves).
>
> We added additional ablations and visualizations in the numerical part. This includes ablations for the number of used features (Table 4 and 5 in the appendix) and a comparison where we just apply the refinement procedure from Section 3.3 without the generative feature training before. Following the comments of Reviewer 1jSb, we also added a comparison with a kernel ridge regression.
> Additionally, we added the visualization of the generated features for the backprop approach. Here, we can see that the features are not pushed to the axis which indicates that the optimization got stuck in a local minimum.

---

### Review · Reviewer_s5U9 · 2025-02-25

**Summary Of Contributions:**

This work proposes a novel, simple and efficient learning paradigm, named Generative Feature Learning, that learns thin 2-layer neural nets with nonlinear activation that has nonconvex MSE loss. It first reduces the problem to only needing to learn layer 1 weights, by finding out the deterministic relation between layer 1 and layer 2 weights through linear regression. Then, it learns layer 1 weights via learning a generative network that generate layer 1 weights. Through numerical experiments, this learning paradigm is proved to overcome the local minima issue of direct loss optimization, and outperform baselines and other standard training paradigms, on elementary function estimation and UCI datasets.

**Audience:**

Yes

**Broader Impact Concerns:**

N/A to this work.

**Claims And Evidence:**

Yes

**Requested Changes:**

* The very general form of the problem (Equation 1) is only presented at the very beginning of the Introduction, which seems unclear in purpose. Maybe consider discussing the applicability and extensibility of the proposed algorithm to this general form of the problem.
* Typo: in Section 3, ii) 2-Layer Neural Network: $W_lin R^{d+1}$ -> $W_l \in R^{d+1}$.
* The part of solving Equation 6 (exact form of Tikhonov regularization) is discussed a bit too briefly. The problem is less than full rank and the specific solution to Equation 6 chosen may need some clarification even though it’s already built in AD packages.
* Question: What will happen if $z$ is not iid sampled but with some covariance?
* Nit: Notation $\bigotimes N$ lacks definition. Is it common to denote iid $N$ times?
* Nit: UCI database lacks citation.
* Initialization of $z$ in $F(z)$ in the methodology may need more clarification.
* Nit: Bolding/Highlighting the best performance in the tables would be nice for a more clear presentation.

**Strengths And Weaknesses:**

* Strengths:
  * Paper is written clearly and easy to follow, especially the narration of problem reduction, core methodology and presentation of results.
  * The method proposed is novel, not overcomplicated to understand, and numerically proved to learn thin 2-layer networks efficiently over common optimization issues and standard learning procedures. It also considers noisy data.

* Weaknesses:
  * Lacks theoretical guarantee on the accuracy of the algorithm, e.g. convergence, generalization error, probably approximately correctness including sample complexity, etc. More specifically, it could be something like error on $w$ estimation being propagated to $b$ estimation error and thus being propagated to $f$ estimation error. The learning of $w$ is through a generative neural net which makes it harder to analyze theoretically, but could we make simplification assumptions on this component or use existing theories to make the analysis feasible?
  * Method seems to target small $N$ (number of hidden layers) only, but in practice $N$ would not be that small ideally.
  * See Requested Changes for minor issues and low-level feedback.

---

> ### Author Response · Authors · 2025-03-06
> **Response to the Review**
>
> Thank you very much for your review. We reply to each of your comments separately.
>
> > Lacks theoretical guarantee on the accuracy of the algorithm, e.g. convergence, generalization error, probably approximately correctness including sample complexity, etc. More specifically, it could be something like error on $w$ estimation being propagated to $b$ estimation error and thus being propagated to $f$ estimation error. The learning of $w$ is through a generative neural net which makes it harder to analyze theoretically, but could we make simplification assumptions on this component or use existing theories to make the analysis feasible?
>
> We agree that a further investigation of the theoretical properties of GFT is an important extension, which we want to consider in the future. In particular, we are interested to characterize minima of the loss function (8) in the space of probability measures and study their dependence on $N$. However, we feel that this goes beyond the scope of this paper.
>
> > Method seems to target small $N$ (number of hidden layers) only, but in practice $N$ would not be that small ideally.
>
> We added a short motivation in the introduction why one should be interested in sparse architectures. Choosing a small $N$ amounts to encoding the sparsity explicitly instead of requiring that many features/weights are pushed towards zero during the optimization.
> For your convenience, we include the additional text:
> Since the Pareto principle suggests that most real-world systems are driven by a few low-complexity interactions, we are interested in representations (1) with only a few features $w_l$.
> Such an explicit restriction to small $N$ also mitigates overfitting, as seen in sparse neural networks, compressed sensing and feature selection.
>
> > Typo and UCI citation
>
> Thanks for spotting. We fixed it.
>
> > The very general form of the problem (Equation 1) is only presented at the very beginning of the Introduction, which seems unclear in purpose. Maybe consider discussing the applicability and extensibility of the proposed algorithm to this general form of the problem.
>
> Our training procedure is indeed applicable to any architecture of the form (1) and not restricted to specific choices of $\Phi$.
> To emphasize this more, we added several references to the general equation (1) throughout the paper.
>
> > The part of solving Equation 6 (exact form of Tikhonov regularization) is discussed a bit too briefly. The problem is less than full rank and the specific solution to Equation 6 chosen may need some clarification even though it’s already built in AD packages.
>
> For $\epsilon\to0$, the solution of (6) converges to the minimal norm solution of (5). In our context it seems reasonable to prefer weights with small norm.
> After regularization, the problem is full rank (the matrix $A_w^T A_w$ is positive semidefinite, hence $A_w^T A_w+\epsilon I$ is invertible) and the solution of (6) unique.
>
> We added these clarifications in the paper.
>
> > Question: What will happen if $z$ is not iid sampled but with some covariance?
>
> From a theoretical viewpoint, we have to ensure that the empirical distribution of the $z$ approximately matches the latent distribution $\eta$. We followed the the easiest approach to achieve that and sampled the $z$ iid from the latent distribution.
>
> We are aware that there exist other approaches to approximate continuous distributions by samples (like quasi-Monte Carlo rules). However, since generators $G_\theta$ of generative models can become quite irregular, these approaches do usually don't provide a significant advantage in combination with generative models. Therefore, we stick with the iid sampling from the latent distribution.
>
> > Nit: Notation $\otimes N$ lacks definition. Is it common to denote iid $N$ times?
>
> Yes, $\mu^{\otimes N}$ denotes $N$-times the product measure of $\mu$. That is, $z\sim\mu^{\otimes N}$ means we take $N$ iid samples $(z_1,...,z_N)$ of $\mu$. We added this definition.
>
> > Initialization of $z$ in $F(z)$ in the methodology may need more clarification.
>
> We added a more extensive explanation in Section 3.3 now: Since these $w^0$ are only an approximation, we refine them similarly as described for the plain feature optimization approach from Section 3.1.
> More precisely, starting in $z^0$ sampled from a fixed distribution, we start with $z^0$ sampled from the generative model trained in Section 3.2.
>
> We also stress now in Section 3.1 that initializing with samples from a Gaussian distribution often fails, since $L(w)$ from (7) is non-convex.
> We refer to this as feature optimization (F-Opt).
> Indeed, our comparisons in Section 4 reveal that feature optimization frequently gets stuck in local minima.
>
> > Nit: Bolding/Highlighting
>
> The best performance was already highlighted by a green cell color. We made it additionally bold now. Moreover, we clarified in the captions of each table, that the highlighted value corresponds to the best performance.

---

### Author Response · Authors · 2025-03-06
**General Answer**

We would like to thank all reviewers for the thorough evaluation of our paper. We summarize the major changes below and provide separate answers to each reviewer. In the pdf, we marked changes in blue.

Changes:

- We changed the acronym GFT-p to GFT-r since the postprocessing was called refinement in Section 3.3.

- We added more comparisons (kernel ridge regression) and ablations (different number $N$ of features, apply refinement without generative model, etc.).

- We added the gradient-based optimization of the neural network to Figure 1.

- We fixed several smaller issues and improved the writing of the paper based on the specific comments of each reviewer.

---

### Decision · Action_Editor_QeZe · 2025-06-27

**Recommendation:** Accept with minor revision

**Comment:**

This paper combines a few ideas into a single method: a generative model of the network's weights, implicit differentiation through a least-squares solver. It's reminiscent of work found in Bayesian neural networks, hypernetworks, meta-learning (like MAML), and backpropagation through fixed-point solvers. The particular approach found in this work is limited to a very restricted class of networks: namely 2 layers, a small number of examples, small number of features, and squared error for the loss, however the paper is up-front about these limitations.

The perspective I find most intriguing is the view that the generative model learns a preconditioner for the neural network learning algorithm. It's not obvious that this should be easier than learning the network itself. For example, the benefits could be due to added noise that is lost when we switch from the mini-batch to the full-batch setting, where naive gradient descent can get stuck in spurious local minima.

It's also unclear the degree to which solving the last layer weights is necessary or complementary to the generative network/preconditioning.

I would therefore suggest two ablations:

1. Train a standard neural network in the full-batch setting with gradient noise, using something like stochastic gradient Langevin dynamics (SGLD) (Welling and Teh, 2011)
2. Train the generative network (sampling z), but learning the output weights b normally instead of solving for their optimal value.

And in general, I think exploring the relative difficulty of learning preconditioners for networks vs learning networks to be a fascinating direction. If we think about a classic smoothing technique like weight decay, this smooths the learning process, but introduces bias in the optimal solution. Preconditioning can be seen as smoothing the optimization landscape itself, which leaves the solution intact. It reminds me of the mystery of distillation: training a large network and then training a small network via distillation from the larger network is somehow easier than training the small network directly.

So overall, despite the limitations of the setting, the small-scale experiments, and limited theory, the paper demonstrates an effective approach and gets at a set of deeper questions that should be of great interest to the community.

**Audience:**

Yes, the learned initialization is similar to ideas in meta-learning and hyper-networks, and this particular formulation could find interest with that audience. The connection to pre-conditioning is itself quite interesting.

**Claims And Evidence:**

Yes, the claims of improved optimization on thin networks with small datasets is verified numerically, although there is not a formal theoretical guarantee of improvement.

---

> ### Author Response · Authors · 2025-07-24
>
> Many thanks for the additional suggestions. We uploaded the camera-ready version and included the additional ablations in Appendix B. Note that we slightly modified your second suggestion: Since the features $w_l$ are sampled from our generative model, there is no persistent correspondence between a feature $w_l$ to the weight $b_l$. Consequently, learning the weights $b$ by a gradient-based optimizer cannot work. Instead, to demonstrate that using the optimal output weights is necessary for GFT, we sample the pairs $(w_l,b_l)$ from a joint generative model.